# Endometrial Cancer in Pre-Menopausal Women and Younger: Risk Factors and Outcome

**DOI:** 10.3390/ijerph19159059

**Published:** 2022-07-25

**Authors:** Nurliza Abdol Manap, Beng Kwang Ng, Su Ee Phon, Abdul Kadir Abdul Karim, Pei Shan Lim, Maimunah Fadhil

**Affiliations:** 1Department of Obstetrics and Gynaecology, Faculty of Medicine, Universiti Kebangsaan Malaysia, Jalan Yaacob Latiff, Bandar Tun Razak, Cheras, Kuala Lumpur 5600, Malaysia; nurlizaabdolmanap@gmail.com (N.A.M.); sephon88@yahoo.com (S.E.P.); abdulkadirabdulkarim@yahoo.com (A.K.A.K.); peishan9900@yahoo.com (P.S.L.); 2Department of Pathology, Hospital Melaka, Jalan Mufti Haji Khalil, Melaka 75400, Malaysia; drmaif2005@yahoo.com

**Keywords:** risk factors, outcome, endometrial cancer, young women, premenopausal

## Abstract

Endometrial cancer is the sixth most common malignancy in women, and it is known to be a disease among postmenopausal women, but there is rising in the number of endometrial cancers among premenopausal women. This study aims to determine the clinical characteristic, risk factors, outcomes, and survival in pre and postmenopausal women with endometrial cancer in Malaysia. A retrospective study was conducted in Hospital Melaka that involved all women who were diagnosed with endometrial cancer in Hospital Melaka from January 2002 until July 2020. All subjects’ histopathological examination result was confirmed, and their clinical data were extracted and transferred into a standardized data checklist and analysed. A total number of 392 cases was obtained from the Annual Cancer Registry Hospital Melaka. However, only 281 cases were studied, including 44.8% premenopausal and 55.2% postmenopausal women. In the premenopausal group, there were higher incidence of obesity (30.8 + 8.6 vs. 28.9 + 7.1), younger age at menarche (12.7 + 1.5 vs. 13.3 + 1.6), lesser parity (1.47 vs. 3.26), and a higher number of nulliparous women (46.8% vs. 19.4%) as compared to postmenopausal group. The premenopausal group tends to be presented with a well-differentiated grading of tumour (52.4%) and a higher incidence of having concomitant endometrial hyperplasia (41.3%). The mean survival among the premenopausal group (200.3 + 7.9 months) is higher compared to postmenopausal group (153.9 + 6.5 months). These findings correlate with good survival and prognosis among the premenopausal group compared to the postmenopausal group.

## 1. Introduction

Endometrial cancer is the fifteenth most common malignancy worldwide, and it is the sixth most commonly occurring malignancy in women [1]. The age-standardised incidence rate (ASR) for uterine malignancy for the year 2012–2016 is 4.6 in 100,000 in the Malaysian population. The lifetime risk for uterine malignancy is 1 in 224 for all females, according to the statistics from Malaysian National Cancer Registry [2].

Endometrial cancer is mainly a disease that occurs primarily in postmenopausal women, with 75–80% of women being postmenopausal at the time of diagnosis [3,4,5]. The incidence of endometrial cancer among premenopausal women varies in the different studies depending on the cut-off age used or evaluation of the ovarian activity [6,7,8,9]. The average menopausal age for the Malaysian population is around 51 years old [10]. The Malaysian National Cancer Registry reported that the majority of women with uterine malignancy are between the ages of 55 and 64 years [2]. The incidence of endometrial cancer in the premenopausal group is between 14 and 20% [4,11,12,13]. It is relatively uncommon (5%) in a patient younger than 40 years old [4,11,12,13].

Most of the patients diagnosed with endometrial cancer presented with postmenopausal bleeding or abnormal uterine bleeding [6,11,14]. Diagnosis of endometrial cancer was made based on HPE of the endometrial tissue biopsy via pipelle sampling or hysteroscopy diagnostic dilatation and curettage (DD & C). Several studies reported an increased risk of endometrial cancer among women who are nulliparous, with early age at menarche or late age of menopause, history of anovulatory cycles, obese, a known case of diabetes mellitus, or having an unopposed oestrogen therapy [5,15,16,17,18,19].

Staging of the disease is important in determining the prognostic survival of each patient. The Federation of Gynaecology and Obstetrics (FIGO) is used to stage endometrial cancer. Another prognostic factor is the histological grading of the tumour. A tumour with better differentiation has better prognostic value. As the tumour loses its differentiation, the prognostic survival reduces [9,20]. The histologic subtype of the endometrial cell is also a major determinant of the prognosis. Endometrioid adenocarcinoma has a better prognosis as compared to clear cell carcinoma, papillary serous carcinoma, and other poorly differentiated carcinomas [2,20,21,22]. The absence of lympho-vascular space invasion (LVSI) is also one of the prognostically favourable histologic types. LVSI signifies the presence of tumour cells inside the endometrium-lined channel, which correlates to lymphatic tumour metastasis [23].

Younger women with endometrial cancer tend to present with an early stage of disease with a favourable histologic type [24]. Thus, conservative management with progestin therapy offers a good alternative option for a woman to preserve their fertility and ovarian function [22,24]. However, conservative management requires further evaluation and discussion to evaluate their effectiveness, safety, risk of recurrence, and overall survival.

There is an increasing number of premenopausal women diagnosed with endometrial cancer each year. Data from the annual cancer registry from the Department of Obstetrics and Gynaecology, Hospital Melaka, showed a higher percentage of young women diagnosed with endometrial cancer (30% cases below the age of 50 years old in 2019, and 16% of them are below 40 years old) compared to local or global statistics. Due to concern regarding the rising number of endometrial cancer and curiosity about the factors contributing to the early development of the disease among the younger women, this study aims to determine the clinical characteristic, risk factors, outcomes, and survival in pre and postmenopausal women with Endometrial cancer in Malaysia.

## 2. Materials and Methods

### 2.1. Study Design

This study involves all women diagnosed with endometrial cancer from January 2002 to September 2020 at Hospital Melaka. The list of the subjects was retrieved from the Hospital Melaka cancer registry. Data from the hospital records and histopathological records were extracted and transferred into a structured standardised data checklist, which was subsequently transferred to an electronic data checklist for analysis. Subjects with incomplete or missing data were excluded from the study. Subsequently, during the analysis, subjects were categorized into two groups, which were premenopausal and postmenopausal. Subjects were classified as premenopausal if, at the time of diagnosis, they did not fulfil the definition of menopause. Menopause is defined as an absence of menses for 12 months due to failure of ovarian function or cessation of menses due to surgical removal of ovaries. All the clinical characteristics and the post-treatment outcome were compared between the two groups.

### 2.2. Instruments

A standardised data checklist used in the study had 5 parts comprising demographic data, a risk factor for endometrial carcinoma, histological information, primary treatment, and outcome.

#### 2.2.1. Demographic Data and Clinical Assessment before a Diagnosis

The first part comprises data on the patient’s demographic and clinical assessment before the diagnosis. These include the patient’s age at diagnosis, ethnicity, menopausal status, duration of the follow-up, clinical presentation, and ultrasound assessment of endometrial thickness before endometrial sampling or intervention.

#### 2.2.2. Risk Factors for Endometrial Carcinoma

The second part comprises questions to evaluate the risk factors for endometrial carcinoma that subjects might have. This includes the age of menarche, age of menopause, number of children, weight and height to measure body mass index (BMI), subject’s comorbidities and history of premalignant lesion, subject’s personal history of malignancy, and first-degree relative with ovarian, endometrial, breast, or colorectal malignancy.

#### 2.2.3. Histological Examination Finding

The third part of the checklist contained histological examination information such as histology subtype, stage of disease at the time of diagnosis, and tumour grading.

#### 2.2.4. Primary Treatment

The fourth part comprises information about the primary treatment received by the subjects, which includes the type of surgery and type and duration of radiotherapy or chemotherapy.

#### 2.2.5. Outcome Post-Primary Treatment

The final part consists of questions to evaluate the outcome post primary treatment. The main goal is to calculate the duration of the progression-free survival and overall survival.

### 2.3. Data Analysis

The sample size for this study was determined using the Cochran formula. By using the prevalence of endometrial cancer among the premenstrual women by Pamela et al. [5] as 12.3%, at 95% confidence interval and significance level of 5%, the minimum calculated sample size was 162 cases.

The Statistical Package of Social Sciences (SPSS) Version 22.0 (IBM Corp., Armonk, NY, USA) was used to analyse the study data. Data are presented as mean (standard deviation, SD) or number, n (percentage, %), for continuous and categorical data, respectively. The demographic data, risk factors, histological examination findings, choice of treatment offered, and post-treatment outcome are compared between the premenopausal and postmenopausal groups.

Chi-square test and Fisher’s exact test were used to look for a significant means difference between the premenopausal and postmenopausal group when analysing the categorical data such as races, presenting complaint, or presence of comorbidity. *T*-test and Mann–Whitney test were used to look for a significant means difference between the two groups when analysing the continuous data, such as duration of symptoms, endometrial thickness during ultrasound examination, or body mass index. A *p*-value of less than 0.05 was interpreted as statistically significant.

The primary outcomes of the study, which are progression-free survival and overall survival, were calculated and the comparison between the premenopausal and postmenopausal group was interpreted using Kaplan–Meier survival analysis.

### 2.4. Ethical Consideration

This comparative, retrospective study had been approved by the Medical Research and Ethics Committee of the Malaysian Ministry of Health (NMRR-20-2143-56407).

## 3. Results

According to data from the Annual Cancer Registry of Hospital Melaka, the total number of women diagnosed with Endometrial carcinoma between January 2002 to September 2020 was 392 cases. However, 87 files went missing, and 16 cases were excluded due to the wrong diagnosis after reviewing the histopathological examination finding. Among 289 files retrieved from the medical record unit of Hospital Melaka, eight cases were excluded due to incomplete data. Therefore, the total number of cases studied was 281 cases, of which 126 women (44.8%) were among the premenopausal group, while 155 women (55.2%) were among the postmenopausal group.

Table 1 demonstrates the clinical characteristics of patients with endometrial cancer cases in Hospital Melaka. The mean age of diagnosis among the premenopausal group is 44.9 + 7.5 years old, which is significantly younger as compared to the postmenopausal group at 62.3 + 6.5 years old. About two-thirds of the cases in both groups were among the Malay ethnic, while less than 2% of each group were among foreigners coming from Indonesia, Philippines, Portugal, and India. The majority of cases presented with abnormal uterine bleeding, which comprised 90.5% among the premenopausal group and 92.9% among the postmenopausal group.

The premenopausal group tends to have a longer duration of symptoms before medical consultation as compared to the postmenopausal group, with a median (IQR) of 12 (6–48) weeks and 8 (4–28) weeks, respectively. The duration of time needed by the doctor to make a diagnosis of endometrial cancer in a premenopausal group was also longer as compared to the postmenopausal group, as represented in median (IQR) as 3.7 (1.1–8.4) weeks and 1.8 (0.5–6.0) weeks, respectively.

The premenopausal group had a higher BMI, attained menarche earlier, and had significantly more nulliparous women, while the postmenopausal group had a higher number of parity and higher number of women with a hypertensive disorder. There was no significant difference in terms of number of cases with diabetes mellitus, measurement of endometrial thickness by ultrasound, number of cases with a previous personal history of breast or colorectal malignancy, and history of a first-degree relative with breast, gynaecological, or colorectal malignancy between two groups. The number of hormonal replacement therapy users, combined oral contraceptive users, tamoxifen users, and smokers were very small in both groups; therefore, it was not further analysed.

Table 2 demonstrates the clinicopathological characteristics and comparison between the two groups. The majority of cases were endometrioid-type endometrial cancer, and only one woman in premenopausal group had non-endometrioid type, which was a clear cell type. There were 11% of women in the postmenopausal group who had non-endometrioid type, which included clear cell carcinoma (2.1%), malignant Mullerian mixed tumour (1.4%), serous type (1.1%), and 3.3% of another non-endometrioid type, such as mucinous adenocarcinoma and giant cell carcinoma.

The majority of endometrial cancer patients in this study were discovered in early disease: more than two-thirds of the patients were classified under stage 1 for both premenopausal and postmenopausal groups. The analysis revealed that the endometrial carcinoma in the premenopausal group was more well-differentiated and had concomitant endometrial hyperplasia at the time of diagnosis. However, there was no significant difference in terms of disease staging, degree of myometrial invasion, lympho-vascular invasion, and presence of synchronous ovarian malignancy.

Among 281 cases of endometrial cancer studied, 93.5% underwent hysterectomy, and 92% of them had bilateral salphingo-opherectomy (BSO). There were only 22% of cases that benefited from pelvic lymph node dissection. Women in the postmenopausal group who had a radical trachelectomy done together with pelvic lymph node dissection had a previous history of subtotal hysterectomy and bilateral salphingo-opherectomy performed many years before recent surgery due to benign disease. The various type of surgeries performed is well-demonstrated in Figure 1.

All stage 1 endometrial cancer patients that had surgery were subsequently classified into four groups based on the Gynaecologic Oncology Group (GOG) criteria: (1) low-risk (LR); (2) low–intermediate-risk (LIR); (3) high–intermediate-risk (HIR); and (4) high-risk (HR) (refer to Appendix A). The classification was made based on the patient’s age, degree of myometrial invasion, tumour grading, and whether there was presence of lympho-vascular stromal invasion during the histopathological examination.

Table 3 shows a comparison of the type of adjuvant treatment received and recurrences by the Gynaecology Oncology Group (GOG) group among the premenopausal women with stage 1 endometrial cancer. None of the premenopausal women who received the adjuvant therapy developed recurrence regardless of the type of adjuvant received or among any Gynaecology Oncology Group (GOG) group.

Among postmenopausal women (Table 4), the percentage of recurrence was higher in the group that received adjuvant therapy, especially in the low–intermediate group that received combined EBRT brachytherapy (33.3%) and in the high-risk group that received brachytherapy or chemotherapy alone (33.3%). However, among the postmenopausal women in the high-risk group, the mean survival time was better in the group that received adjuvant therapy compared to the group that did not receive any adjuvant (138.9 + 21.6 months vs. 67.4 + 40.4 months), but the difference was not significant, with *p* = 0.498, as demonstrated in Table 5.

Figure 1 demonstrates a survival analysis of a total of 158 women who were diagnosed with endometrial cancer from January 2002 until September 2015, showing a 5 year-survival rate of 100% for the premenopausal group but only 88% for the postmenopausal group. The mean survival time among the premenopausal group is higher, at (200.3 + 7.9) months, compared to postmenopausal group’s (153.9 + 6.5) months, and it is statistically significant with *p* = 0.014.

Figure 2 illustrated the survival curve in terms of progression-free survival among endometrial cancer patients who were diagnosed from January 2002 until September 2015. According to the Kaplan–Meier survival analysis (Figure 2), the premenopausal group had better progression-free survival than the postmenopausal group, and the difference was statistically significant (*p* = 0.032). From the survival curve, there was about 1.5% among the premenopausal group, and 6% among the postmenopausal group developed recurrence within 1 year, while in 3 years, 3% of premenopausal group and 10% among postmenopausal group developed recurrence. The recurrence rate among premenopausal group in 5 years remained unchanged, but in the postmenopausal group, it was increased to 12%.

## 4. Discussion

Endometrial cancer was used to be thought of as a disease of older women and among the postmenopausal group. Previously, the majority of cases were diagnosed at the sixth or seventh decades of life [5]. However, this is different from our findings, in which 44.8% of cases were among the premenopausal group, and the mean age of diagnosis in this group was 44.9. The number of premenopausal women diagnosed with endometrial cancer is about two times higher in our study (44.8%) compared to the previous study by Takehiko et al. (24.2%) and Laure Dossus et al. (19.5%) [12,15]. Another local study by Wan Adnan et al. in 2016 reported their endometrial cancer cases among premenopausal groups were as high as 32.4% [25].

Despite this, according to the Malaysian National Cancer Registry Report 2012–2016, by ethnicity, China had the highest incidence of cancer in general for both sexes [1]. However, the incidence of endometrial cancer had its majority in the Malay population, followed by Chinese and the least from Indian ethnicities, as supported by the result from our finding and a local study by Wan Adnan et al. [25].

Current guidelines and studies consider abnormal uterine bleeding in premenopausal women, especially those aged less than 45 years old, as low risk for endometrial cancer; thus, referral for biopsy appears unwarranted and over-cautious [6]. With the rising number of endometrial cancers among the premenopausal group, we advise a specialist referral and endometrial biopsy to be done in young women with risk factors. One of the limitations of the study was poor documentation and inadequate information retrieved about the nature of the abnormal uterine bleeding from the retrospective sampling. It will be beneficial for the details on the abnormal uterine bleeding to be gathered and analysed for a better understanding of the pattern and characteristics of the uterine bleeding associated with endometrial cancer.

Excess oestrogen, such as in obesity, has been associated with a higher incidence of endometrial cancer [4]. From this study, more premenopausal women with high BMI (obesity), nulliparity, and earlier age to attained menarche were diagnosed to have endometrial cancer. The incidence of concomitant endometrial hyperplasia was also higher in premenopausal as compared to postmenopausal group, which also supports the evidence of prolonged unopposed oestrogen exposure. Obese women with an anovulatory menstrual cycle are at risk of developing diabetes mellitus and endometrial cancer. Data presented in Table 1 show the incidence of diabetes mellitus was slightly lower among the premenopausal group, which is contrary to other previous studies showing the incidence of diabetes was relatively higher among the premenopausal group; however, the difference was not significant [5,12].

Many studies report that young women with endometrial carcinoma tend to have a prognostically favourable histological type, such as endometrioid-type tumour, well-differentiated tumour grading, early staging, less myometrial involvement, and absence of lympho-vascular stromal invasion [11,12,13,25]. This is consistent with the finding in this study, as our premenopausal women tend to present with endometrioid-type tumours and well-differentiated tumour grading as compared to postmenopausal group. However, there was no significant difference in terms of disease staging and lympho-vascular stromal invasion. This could be attributed to the incomplete surgical staging performed, resulting in inaccurate disease staging.

A consensus has been achieved, whereby no adjuvant therapy is recommended in managing low-risk endometrial cancer, as results from multiple randomized trials show no improvement in prognosis [26,27,28,29]. Moreover, patients treated with adjuvant radiotherapy are at risk of developing toxicity, mainly gastrointestinal [30]. In this study, only one case of recurrence was reported in each group (premenopausal (2.9%) and postmenopausal (6.3%)) among the low-risk group, who did not receive any adjuvant therapy. Three patients received adjuvant therapy in the low-risk group; however, any clinical evidence of radiotherapy toxicity was not evaluated in the study, but none of the patients developed any recurrence.

The use of adjuvant radiotherapy in women with intermediate risk help to reduce the risk for loco-regional recurrence. There were about 30% of women at risk of developing recurrence without any additional therapy, especially in older women, with deeper myometrial invasion, higher tumour grade, and presence of lympho-vascular stromal invasion [30,31]. The result in this study supports the literature but was only applicable to the premenopausal group. The rate of recurrence among postmenopausal women in the intermediate-risk group was higher in women that received adjuvant compared to women that did not receive adjuvant therapy. The factors that might contribute to the result obtained were the patient’s age, general well-being, presence of the unfavourable histological finding, and possible incomplete surgical staging. Three out of four postmenopausal women who were classified under the intermediate-risk group, which received pelvic radiotherapy with a brachytherapy boost, were more than 60 years old, obese, and with higher tumour grading, also developing recurrence within the first year of completed treatment. Two out of three of them underwent a simple hysterectomy with bilateral salphingo-opherectomy alone, which did not benefit from gynae-oncology expertise at that time.

The high-risk group is at risk of disease recurrence, and approximately 15–20% of women with high-risk endometrial cancer are at risk of distant metastasis [30]. Previously, pelvic radiotherapy has been the standard treatment to prevent disease recurrence; however, many randomized trials have to use chemotherapy as an alternative or additional therapy to improve survival [32,33,34]. The combination of chemotherapy and EBRT was associated with a 36% reduction in the risk of relapse or death [34]. Two recurrence cases in the high-risk group among the postmenopausal women were both more than 60 years old with a non-endometrioid-type endometrial cancer who received a single type of adjuvant therapy, which was either brachytherapy or chemotherapy alone.

In general, the prognosis and overall survival for premenopausal group are better as compared to postmenopausal group. This is mainly because most of the premenopausal group presented in early disease with favourable histological findings (endometrioid type, well-differentiated tumour), were physically fit, and had fewer comorbidities [11].

There were some limitations in this study. As we mentioned earlier, a large number of missing files (20% of the total number of endometrial cancers reported in the Annual Cancer Registry of Hospital Melaka) may have affected the overall data analysis of the study. In a hospital with a paper-based medical record, missing variables from the torn or damaged pages and incomplete data regarding particularly family and social history further reduced the total number of cases studied. A high number of censored cases in the survival analysis mainly contributed to the high number of patients who defaulted in follow-up; thus, the recurrence or survival rates may be underestimated or overestimated, as we are unsure regarding the patients’ current wellbeing. In addition to that, the use of median survival time is more appropriate and reliable as compared to mean survival time. In this study, the median survival time cannot be computed, as the survival curve did not drop to below 50% of the population, while the use of the mean survival time in this study may underestimate the actual survival and treatment benefit, as the measurement is restricted only to the follow-up period of the clinical study.

## 5. Conclusions

The number of endometrial cancers in premenopausal group is in a rising trend; thus, a high index of suspicion of malignancy is needed in young women presenting with abnormal uterine bleeding associated with conditions that contribute to excessive unopposed oestrogen. However, most of the patients in the premenopausal group usually presented with an early disease with a favourable histological type, and generally, it is associated with a good prognosis and better survival.

## Figures and Tables

**Figure 1 ijerph-19-09059-f001:**
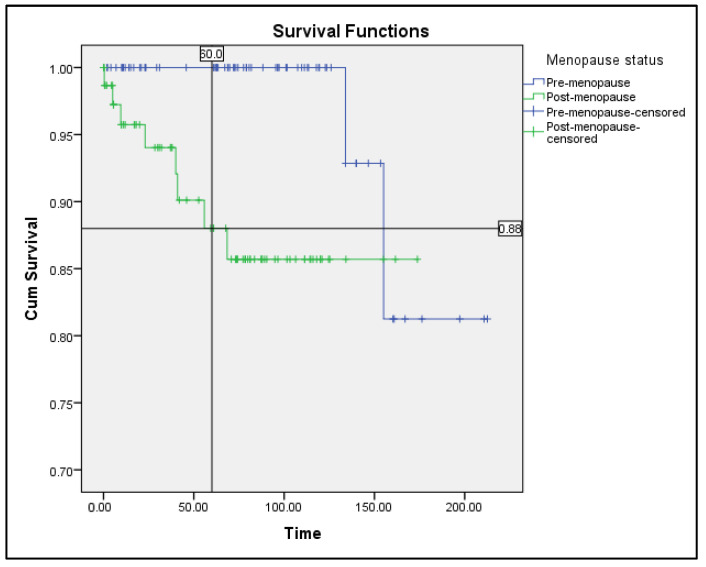
Survival curve (Kaplan–Meier method). Premenopausal group versus postmenopausal group. Overall survival among endometrial cancer patients in Hospital Melaka from 2002–2015.

**Figure 2 ijerph-19-09059-f002:**
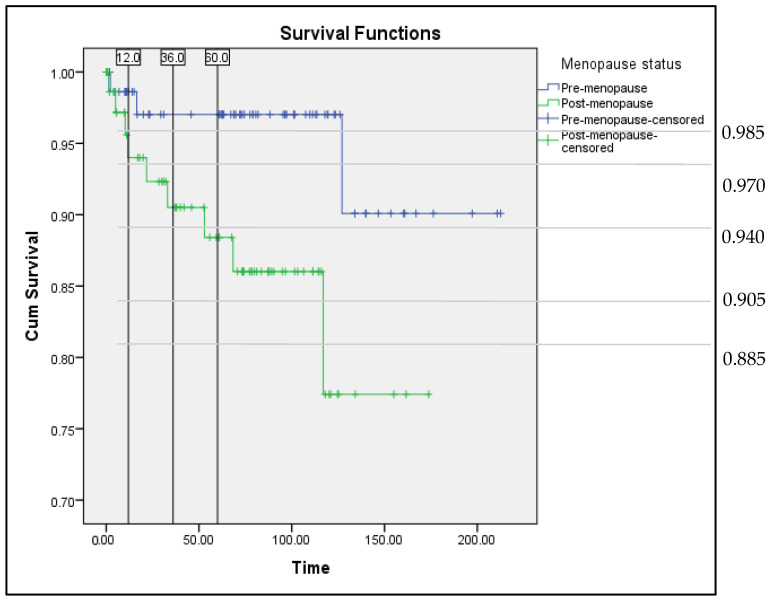
Survival curve (Kaplan–Meier method). Premenopausal group versus postmenopausal group. Progression-free survival among endometrial cancer patients in Hospital Melaka from 2002–2015.

**Table 1 ijerph-19-09059-t001:** Clinical characteristics of Endometrial Cancer cases.

	Premenopause Subgroup (126)	Postmenopause Subgroup (155)	*p*-Value
Age at Diagnosis	44.9 (7.5)	62.3 (6.5)	<0.001
Race			0.622
Malay	89 (70.6%)	99 (63.9%)	
Chinese	26 (20.6%)	38 (24.5%)	
Indian	9 (7.1%)	16 (10.3%)	
Others	2 (1.6%)	2 (1.3%)	
Presentation			0.117
Abnormal uterine bleeding	114 (90.5%)	144 (92.9%)	
Pelvic mass	9 (7.1%)	3 (1.9%)	
Abdominal pain	1 (0.7%)	2 (1.3%)	
Others	2 (1.5%)	5 (3.2%)	
Duration of symptom (week)	12 (6–48)	8 (4–28)	0.003
Duration to diagnose (week)	3.7 (1.1–8.4)	1.8 (0.5–6.0)	0.003
Endometrial thickness (mm)	19.8 (10.8)	18.7 (10.9)	0.518
BMI (kg/m^2^)	30.8 (8.6)	28.9 (7.1)	0.040
Age at menarche	12.7 (1.5)	13.3 (1.6)	0.042
Number of children	1.47 (0–3)	3.26 (1–5)	<0.001
Nulliparity	59 (46.8%)	30 (19.4%)	<0.001
Hypertension	49 (38.9%)	96 (61.9%)	<0.001
Diabetes mellitus	38 (30.2%)	57 (36.8%)	0.213
Personal history of malignancy	4 (3.2%)	12 (7.8%)	0.124
Ovarian	0	0	
Breast	3 (2.4%)	8 (5.2%)	
Colorectal	1 (0.8%)	2 (1.3%)	
Others (Lymphoma)	0	2 (1.3%)	
First-degree relative with malignancy	17 (13.5%)	28 (18.0%)	0.299
Endometrial	4 (3.2%)	9 (5.8%)	
Ovarian	0	1 (0.6%)	
Breast	10 (7.9%)	14 (9.0%)	
Colorectal	5 (3.9%)	7 (4.5%)	
COCP user (all < 5 years)	12 (9.5%)	14 (9.0%)	NA
Smoking	1 (0.8%)	2 (1.3%)	NA
Tamoxifen	0	6 (3.9%)	NA
HRT user	0	3 (1.9%)	NA

BMI, body mass index; COCP, combined oral contraceptive pill; HRT, hormone replacement therapy; NA, not assessed. Values are in mean (SD), median (IQR), n (%).

**Table 2 ijerph-19-09059-t002:** Clinicopathological characteristics of endometrial cancer cases.

		Premenopause Subgroup (126)	Postmenopause Subgroup (155)	*p*-Value
Subtype	Endometrioid	125 (99.2%)	138 (89%)	0.001
	Non-endometrioid	1 (0.8%)	17 (11%)	
	Clear cell	1 (100%)	5 (3.2%)	
	MMT	0	4 (2.6%)	
	Serous	0	3 (1.9%)	
	Others	0	5 (3.3%)	
Tumour grading	Well-differentiated	66 (52.4%)	47 (30.3%)	0.001
	Moderately differentiated	49 (38.9%)	80 (51.6%)	
	Poorly differentiated	6 (4.8%)	13 (8.4%)	
FIGO staging	Stage 1	90 (71.4%)	113 (72.9%)	0.470
	1a	45 (50%)	48 (42.5%)	
	1b	45 (50%)	65 (57.5%)	
	Advanced stage	35 (27.8%)	36 (23.2%)	
	Stage 2	18 (14.3%)	18 (11.6%)	
	Stage 3	15 (11.9%)	14 (9.0%)	
	Stage 4	2 (1.6%)	4 (2.6%)	
LVSI	Presence	23 (18.3%)	29 (18.7%)	0.960
Synchronous ovarian malignancy	Presence	7 (5.6%)	2 (1.3%)	0.085
Concomitant hyperplasia	Yes	52 (41.3%)	43 (27.7%)	0.017

Values are in n (%). MMT, malignant Mullerian mixed tumour.

**Table 3 ijerph-19-09059-t003:** Comparison of adjuvant treatments, and recurrences by Gynaecology Oncology Group (GOG) for Premenopausal group among women with stage 1 endometrial cancer in Hospital Melaka from 2001–2020.

	No Adjuvant		Brachytherapy Alone		EBRT Alone		EBRT + Brachytherapy		CCRT		Chemotherapy Alone		Total
	n (50)	Recur n (%)	n (12)	Recur n (%)	n (1)	Recur n (%)	n (21)	Recur n (%)	n (0)	Recur n (%)	n (0)	Recur n (%)	
LR	34	1 (2.9)	2	0 (0)	0	0 (0)	0	0 (0)	0	0 (0)	0	0 (0)	36
LIR	12	1 (8.3)	3	0 (0)	1	0 (0)	14	0 (0)	0	0 (0)	0	0 (0)	30
HIR	4	0 (0)	7	0 (0)	0	0 (0)	4	0 (0)	0	0 (0)	0	0 (0)	15
HR	0	0 (0)	0	0 (0)	0	0 (0)	3	0 (0)	0	0 (0)	0	0 (0)	3

LR, low-risk; LIR, low–intermediate-risk; HIR, high–intermediate-risk; HR, high-risk. EBRT, external beam radiotherapy; CCRT, combined chemo-radiotherapy.

**Table 4 ijerph-19-09059-t004:** Comparison of adjuvant treatments, and recurrences by the Gynaecology Oncology Group (GOG) for postmenopausal group among women with stage 1 endometrial cancer in Hospital Melaka from 2001–2020.

	No Adjuvant		Brachytherapy Alone		EBRT Alone		EBRT + Brachytherapy		CCRT		Chemotherapy Alone		Total
	n (44)	Recur n (%)	n (28)	Recur n (%)	n (6)	Recur n (%)	n (29)	Recur n (%)	n (0)	Recur n (%)	n (3)	Recur n (%)	
LR	16	1 (6.3)	1	0 (0)	0	0 (0)	0	0 (0)	0	0 (0)	0	0 (0)	17
LIR	15	0 (0)	12	0 (0)	0	0 (0)	6	2 (33.3)	0	0 (0)	0	0 (0)	33
HIR	9	1 (11.1)	12	0 (0)	5	0 (0)	18	2 (11.1)	0	0 (0)	0	0 (0)	44
HR	4	1 (25)	3	1 (33.3)	1	0 (0)	5	0 (0)	0	0 (0)	3	1 (33.3)	16

LR, low-risk; LIR, low–intermediate-risk; HIR, high–intermediate-risk; HR, high-risk. EBRT, external beam radiotherapy; CCRT, combined chemo-radiotherapy.

**Table 5 ijerph-19-09059-t005:** Recurrences and mean survival time by the Gynaecology Oncology Group (GOG) in stage 1 endometrial cancer in Hospital Melaka from 2001–2020.

GOG Risk Type		Premenopausal (84)				Postmenopausal (110)			
		Recurrence (n (%))	Total Cases (n)	Mean Survival Time (Month)	Log Rankp-Value	Recurrence (n (%))	Total Cases (n)	Mean Survival Time (Month)	Log Rankp-Value
LR	No adjuvant	1 (2.9)	34	155.0 ± 5.8	0.758	1 (6.3)	16	84.2 ± 4.7	0.796
	Brachytherapy	0	2	NA		0	1	NA	
LIR/HIR	No adjuvant	1 (6.2%)	16	198.5 ± 13.5	0.188	1 (4.2)	24	150.8 ± 10.3	0.716
	Adjuvant	0	29	NA		4 (7.5)	53	137.2 ± 9.0	
	Brachy	0	10			0	24		
	EBRT	0	1			0	5		
	EBRT + Brachy	0	18			4	24		
HR	No adjuvant	0	0	NA	NA	1 (25)	4	67.4 ± 40.4	0.498
	Adjuvant	0	3	NA		2 (16.7)	12	138.9 ± 21.6	
	Brachy	-	-			1	3		
	EBRT	-	-			0	1		
	EBRT + Brachy	0	3			0	5		
	Chemotherapy	-	-			1	3		

Value for mean survival time: mean ± SD in months. LR, low-risk; LIR, low–intermediate-risk; HIR, high–intermediate risk; HR, high-risk; Brachy, brachytherapy.

## Data Availability

The data presented in this study are available on request from the corresponding author. The data are not publicly available due to ownership belongs to the institution where the study was conducted.

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
