# Peer review of "Endometrial Cancer in Pre-Menopausal Women and Younger: Risk Factors and Outcome"

_ijerph, 2022, doi:10.3390/ijerph19159059_

Round 1

Reviewer 1 Report

No comments

Reviewer 2 Report

1.      What is the fertility preserve policy of pre-menopause women may need to be mentioned.

2.      How the diagnosis of EM cancer was made in pre-menopause women should  be mentioned.

3.      How many the EM cancer accidently noted after operation should be mentioned. 

4.      The prognosis factor and survival may be analyzed over pre-menopause women group. 

5.       The cohort in this study diagnosed with endometrial cancer between 2002-2020. Which version of FIGO or AJCC staging system did you adopt? 

6.      The survival outcome in pre-menopausal women group is better than post-menopausal group. It is possible due to most of them were endometrioid type, and there were more patients diagnosed as grade 1 disease than post-menopausal group. Besides, in pre-menopausal women, there was a lower recurrence rate after adjuvant therapy in stage 1 disease. However, the mean survival time of “stage 1, low risk” patients is much worse than “stage 1, low-intermediate/high intermediate” patients in both pre-menopausal women and post-menopausal women. Is it possible under-estimated or over-estimated due to the missing data? 

Reviewer 3 Report

In their Systematic research “Endometrial Cancer In Pre-Menopausal Women And Younger: Risk Factors And Outcome" the authors, Manap et al. intended to incorporate scientific data and ascertain the clinical characteristics, risk factors, outcomes, and survival in pre and postmenopausal women with endometrial cancer. This constitutes an interesting and broadly coherent body of work.

However, a few substantial issues need to be resolved, as listed below.

1.     Please include a brief statement in the abstract stating that the research was restricted to the Malaysian population, as otherwise, it appears that the figures are based on a global database.

2.     The introduction by the authors contains a variety of approaches, which muddles the main topic of this research project. I would suggest the authors to rewrite the introduction, please.

3.     Please make a flow chart for the study design.

4.     In the result section, please insert the subgroup.

5.     In the method section, the authors have mentioned that they have done histological analysis. Please provide representative Histological pictures.
